# Exploring Novel Genomic Loci and Candidate Genes Associated with Plant Height in Bulgarian Bread Wheat via Multi-Model GWAS

**DOI:** 10.3390/plants13192775

**Published:** 2024-10-03

**Authors:** Tania Kartseva, Vladimir Aleksandrov, Ahmad M. Alqudah, Matías Schierenbeck, Krasimira Tasheva, Andreas Börner, Svetlana Misheva

**Affiliations:** 1Institute of Plant Physiology and Genetics, Bulgarian Academy of Sciences, Acad. G. Bonchev Str., Block 21, 1113 Sofia, Bulgaria; tania_karceva@abv.bg (T.K.); aleksandrov@gbg.bg (V.A.); krasitasheva@abv.bg (K.T.); 2Biological Science Program, Department of Biological and Environmental Sciences, College of Arts and Sciences, Qatar University, Doha P.O. Box 2713, Qatar; aalqudah@qu.edu.qa; 3Leibniz Institute of Plant Genetics and Crop Plant Research (IPK Gatersleben), Corrensstraße 3, 06466 Seeland, OT Gatersleben, Germany; schierenbeck@ipk-gatersleben.de (M.S.); boerner@ipk-gatersleben.de (A.B.); 4CONICET CCT La Plata, 8 n°1467, La Plata 1900, Argentina

**Keywords:** gene expression, genome-wide association scan, grain yield, green revolution, plant stature, *Rht* genes, semi-dwarfing genes, *Triticum aestivum* L.

## Abstract

In the context of crop breeding, plant height (PH) plays a pivotal role in determining straw and grain yield. Although extensive research has explored the genetic control of PH in wheat, there remains an opportunity for further advancements by integrating genomics with growth-related phenomics. Our study utilizes the latest genome-wide association scan (GWAS) techniques to unravel the genetic basis of temporal variation in PH across 179 Bulgarian bread wheat accessions, including landraces, tall historical, and semi-dwarf modern varieties. A GWAS was performed with phenotypic data from three growing seasons, the calculated best linear unbiased estimators, and the leveraging genotypic information from the 25K Infinium iSelect array, using three statistical methods (MLM, FarmCPU, and BLINK). Twenty-five quantitative trait loci (QTL) associated with PH were identified across fourteen chromosomes, encompassing 21 environmentally stable quantitative trait nucleotides (QTNs), and four haplotype blocks. Certain loci (17) on chromosomes 1A, 1B, 1D, 2A, 2D, 3A, 3B, 4A, 5B, 5D, and 6A remain unlinked to any known *Rht* (*R*educed *h*eigh*t*) genes, QTL, or GWAS loci associated with PH, and represent novel regions of potential breeding significance. Notably, these loci exhibit varying effects on PH, contribute significantly to natural variance, and are expressed during seedling to reproductive stages. The haplotype block on chromosome 6A contains five QTN loci associated with reduced height and two loci promoting height. This configuration suggests a substantial impact on natural variation and holds promise for accurate marker-assisted selection. The potentially novel genomic regions harbor putative candidate gene coding for glutamine synthetase, gibberellin 2-oxidase, auxin response factor, ethylene-responsive transcription factor, and nitric oxide synthase; cell cycle-related genes, encoding cyclin, regulator of chromosome condensation (RCC1) protein, katanin p60 ATPase-containing subunit, and expansins; genes implicated in stem mechanical strength and defense mechanisms, as well as gene regulators such as transcription factors and protein kinases. These findings enrich the pool of semi-dwarfing gene resources, providing the potential to further optimize PH, improve lodging resistance, and achieve higher grain yields in bread wheat.

## 1. Introduction

In the 21st century, food security is a critical concern due to the rapid growth of the global population, limited arable land, and climate challenges. Bread wheat (*Triticum aestivum* L.) serves as a staple food for approximately half of the population, providing around 20% of daily calories and protein. Increasing wheat production quantitatively and qualitatively is essential to meet future food demands. In Bulgaria, bread wheat is the major cereal crop, representing 28% of the total agricultural production, and about 66% of the cereal production for 2022. Agricultural land under bread wheat accounts for about 1.2 million hectares, the average annual production spans from 4.7 to 7.1 million tonnes, and the average grain yield is 3.9–5.9 tonnes/hectare (data for the last 10 years according to Annual Reports of the Bulgarian Ministry of Agriculture, Food and Forestry; https://www.mzh.government.bg/en/policies-and-programs/reports/agricultural-report/; accessed on 22 September 2024).

The spreading of “Green Revolution” semi-dwarf varieties, carriers of *Rht* (*R*educed *h*eigh*t*) genes, underscored the importance of managing plant height (PH) to achieve higher grain yields. Unlike wild-type plants, which grow tall in response to abundant fertilization and irrigation, semi-dwarf varieties exhibit shorter stature [1]. These short-stem plants allocate a greater proportion of their biomass to grain rather than foliage, resulting in an improved harvest index [2]. Additionally, their reduced height makes them more resistant to lodging caused by wind and rain and increases the grain metabolic source. Furthermore, semi-dwarf varieties respond favorably to nitrogen applications, enhancing grain yield while minimizing straw biomass. Consequently, these varieties significantly contributed to the remarkable increase in cereal crop yields worldwide [3,4]. However, the *Rht* genes are associated with shorter coleoptiles, reduced seedling emergence and early growth [5,6,7], negative effects on floral traits [1], reduced nitrogen-use efficiency (NUE) [8], and lower grain yield than taller plants in warm and dry environments [9,10].

The widely adopted “Green Revolution” genes *Rht-B1b* (*Rht1*) and *Rht-D1b* (*Rht*2) result from knock-out mutations in homoeoloci on group 4 chromosomes, specifically encoding DELLA proteins (DELLAs). DELLAs play a central role as regulators in the gibberellin (GA) phytohormone signaling pathway, suppressing GA-responsive growth by physically interacting with transcription factors and other downstream components [11]. In warmer climate zones, such as Southeastern Europe and large parts of China, the alternative *Rht8* gene on chromosome 2D is widely used [7,12,13,14,15,16,17]. The *Rht8* semi-dwarf phenotype has been attributed to reduced cell elongation, resulting in shorter stem internodes, possibly due to decreased sensitivity to brassinosteroids [18]. Recent studies revealed that the *Rht8* gene encodes a protein with a zinc finger BED-type motif and an RNase H-like domain that is essential for controlling bioactive GA synthesis and, consequently, stem height [19,20]. The portfolio of semi-dwarfing genes with commercial relevance in wheat breeding has recently expanded to include a new gene, *Rht24*, located on chromosome 6A [21]. This gene has been actively selected by breeders over the last few decades, showing no detrimental effects on floral traits related to hybrid production (anther extrusion) as well as resistance against Fusarium head blight [21,22]. *Rht24* encodes GA 2-oxidase, a key enzyme in the GA biosynthesis pathway. The dwarfing effect of the mutant allele *Rht24b* is attributed to increased expression of GA 2-oxidase and a lower content of bioactive GA in stems [23].

Additionally, researchers have named and cataloged 22 other dwarfing genes, totaling 26 to date [23,24,25,26], and a number of *Rht* genes have been cloned [27]. Furthermore, over 600 genomic loci have been reported to influence PH based on quantitative trait loci (QTL) linkage analyses or genome-wide association scan (GWAS) assays that have been published by 2023 (summarized in [27]). These PH-related genomic regions span all 21 wheat chromosomes, with 65 QTL-rich clusters identified, particularly on chromosomes 2A, 2B, 3A, 4B, 5A, and 5B [27].

Wheat breeders continue working to adjust PH and optimize apical dominance, promote biomass production, improve lodging resistance, and facilitate mechanical harvesting, all with the ultimate goal of enhancing crop productivity. Achieving the optimum PH involves complex genetic and molecular pathways related primarily to phytohormones’ biosynthesis and signal transduction, but also to cell division, and other growth-related processes. Understanding these mechanisms can help breeders develop high-yielding wheat varieties that are less prone to lodging.

In light of the pleiotropic effects resulting from mutations in height-controlling genes, and in view of projected resource limitations, achieving a wheat plant with longer coleoptiles, improved NUE, drought tolerance, and higher grain yield will require continuous optimization of PH [27]. The abundance of reported height-controlling loci provides a basis for fine-tuning the proposed ideotype of plant stature. Identifying novel trait-related loci could benefit PH regulation in different environments and resource availability. In addition, the effectiveness and success of genomics-assisted wheat breeding in achieving future genetic gains depends on various factors, among which the availability of well-characterized germplasm collections that encompass the global wheat biodiversity is crucial [28].

Bulgaria has a long history of successful wheat breeding, and its initiatives have produced important advancements in agriculture. The first varieties were created by relatively uniform selections performed within landraces at the turn of the 20th century, followed by crosses involving local and foreign accessions. The first landmark of the genetic gains for the Bulgarian wheat germplasm was the introduction of semi-dwarfing genes in the 1960s, predominantly *Rht8*, marking the beginning of a science-based wheat breeding program. During the 1990s, wheat breeding shifted its focus to introduce *Rht-B1* alleles [15], and to include varieties with better grain end-use quality in addition to those that are highly productive. The Dobrudzha Agricultural Institute (Danube Plain, Northern Bulgaria), the Institute of Plant Genetic Resources (Thracian Lowland, Southern Bulgaria), and the Institute of Genetics (now Institute of Plant Physiology and Genetics; Western Bulgaria) have released over 140 varieties since 1960. Private enterprises have also launched successful varieties since the 1990s.

In our study, we applied a GWAS using a 25K Infinium iSelect array to analyze PH across different environments in 179 bread wheat genotypes, including advanced contemporary varieties, historical varieties, and landraces from Bulgaria. Three computational approaches were used: the mixed linear model (MLM), Fixed and random model Circulating Probability Unification (FarmCPU), and Bayesian-information and Linkage-disequilibrium Iteratively Nested Keyway (BLINK). Novel promising associations were identified in more than one environment or by more than one method. Loci associated with reduction or increase in PH were distinguished, and putative candidate genes with a role in plant growth, and high expression levels were determined. Unraveling the phenotype–genotype relationship for PH provides a valuable genetic resource for optimizing plant stature through marker-aided selection methods.

## 2. Results

### 2.1. Phenotypic Data Analysis

Plant height was recorded as the distance between the base of the stem to the ear top, awns excluded, in three crop seasons (field environments, years)—2013/14, 2016/17, and 2020/21, denoted hereafter by the year of harvest. PH varied broadly in the individual growing seasons and throughout those seasons, with an average of 103.2 cm across the years (Table 1 and Appendix A, Figure 1). The PH mean values in the individual years were significantly different at *p* ≤ 0.001. The highest mean height was reached in 2014 (34% and 22% higher mean values compared to 2017 and 2021, respectively), which can be explained by the prolonged period of high rainfall spanning from stem elongation to the grain filling stage (Appendix A). In 2014, PH ranged from 89.4 to 170.0 cm, representing a 1.9-fold difference. In 2017 and 2021, although the plants were significantly shorter, the range of trait values represented a 2.3- and 2.2-fold difference, respectively. The coefficient of variation was highest in 2017 (22.1%). To account for the influence of the field environment, the best linear unbiased estimator (BLUE) values were calculated for each accession, assuming genotype as a fixed factor and year as a random effect. BLUEs varied across the years from 82.6 to 140.5 cm, on average 103.2 cm (Table 1).

In our study, we observed a substantial decrease in PH (*p* ≤ 0.001) when comparing accessions grown or developed before the 1960s with those released after that period (Appendix A). Interestingly, there was no discernible difference between the group of varieties released during the period 1961–1990 and those released after 1990 (Appendix A). Earlier studies on *Rht* gene distribution among Bulgarian bread wheat accessions showed a prevalence of *Rht8*, either alone or in combination with *Rht-B1b/d*, and a small number of varieties with only *Rht-B1* alleles [7,15]. The genotypes carrying *Rht8*, *Rht-B1b/d + Rht8*, or *Rht-B1b/d* alleles did not differ in BLUEs averaged across the crop seasons. However, they differed significantly (*p* ≤ 0.001) from the subset of accessions with wild alleles at the *Rht-B1* and *Rht-D1* loci, and alleles different from the *Xgwm261-192bp* allele, which is diagnostic for *Rht8* [12] (Appendix A).

In all environments, the phenotypic data for PH exhibited a normal distribution based on the Shapiro–Wilk test (*p* ≤ 0.05), exhibiting two distinct peaks in each environment, representing the primary groups of more recent semi-dwarf varieties and the old germplasm (tall landraces and historical accessions) (Figure 1). Factorial analysis of variance (ANOVA) revealed significant effects of genotype, environment, and their interactions, explaining the broad phenotypic variation for PH within the association panel (Appendix A). The broad-sense heritability (*h*^2^) values for the trait were moderate to high ranging from 0.77 to 0.97 in the individual years, and 0.98 across the years (Table 1). Moderate to high positive Pearson’s correlation coefficients (*r*) over the years and with the BLUEs (ranging from 0.71 to 0.95) were calculated for PH (Appendix A).

### 2.2. GWAS Results

The association mapping analysis involved 17,083 high-quality polymorphic SNPs and datasets for PH in 179 wheat accessions across three field environments. Genotypic data were obtained from the 25K Infinium iSelect array (SGS Institut Fresenius GmbH TraitGenetics Section, Gatersleben, Germany), as described in a previous work [29]. Although the trait exhibited moderate to high broad-sense heritability estimates in each crop season (Table 1), ANOVA revealed significant genotype-by-environment interactions (Appendix A). Consequently, we conducted association analysis using phenotypic data from the individual years, followed by analysis based on BLUE values across the years. For GWAS, three methods were employed: the classical single-locus MLM and two multivariate methods—FarmCPU and BLINK. MLM accounted for the previously established population structure (Q) [29] and relatedness among individuals using a kinship matrix (K) [30].

Based on the quantile–quantile (Q–Q) plots, the MLM (Q + K) model displayed a significant deviation of the observed *p*-values from the expected distribution, suggesting potential overestimation of positive genomic signals in specific cases (Appendix A). To mitigate false positive associations, we established a significance criterion of −log_10_ (*p*-value) > FDR (False Discovery Rate), with *p* < 0.001, tailored for each environment. Interestingly, the Q–Q plots for both multivariate models trended downward across some environments, indicating that FarmCPU and BLINK are prone to reporting false negatives (Appendix A). Notably, however, the BLINK model exhibited markedly inflated *p*-values in 2021 and with BLUEs.

In total, we identified 259 significant QTNs for PH, distributed across all chromosomes except 4D. Of these, 87 were detected using the MLM Q + K model (−log_10_ (*p*-value) > FDR) at an adjusted significance level of adjusted *p*-value < 0.001, while 172 were identified using the multivariate methods with an adjusted *p*-value (FDR) < 0.01 (Appendix A). The MLM primarily detected markers associated with PH in 2014 and 2017 (Figure 2a), whereas BLINK predominantly recorded marker associations in 2021 and with BLUEs. FarmCPU yielded only four significant QTNs in 2014 (Figure 2b).

The majority of the meaningful QTNs were identified in a single environment. However, some shared QTNs were found: (1) 17 QTNs were identified with BLUEs and in one growing season, almost exclusively with BLINK, (2) 13 QTNs were identified in two growing seasons, predominantly with MLM, and (3) 2 QTNs were identified with more than one method and growing season (Table 2, Figure 2). These QTNs can be considered stable.

Clusters of adjacent significant SNPs were found on chromosomes 1B, 3A, and 6A. For these chromosome regions, we constructed haplotype blocks (haploblocks) based on linkage disequilibrium (LD) information for the individual chromosomes [30], and the highest pairwise LD (*r*^2^) values between markers within these regions. Specifically, we defined one haploblock on each of chromosomes 1B and 6A, and two haploblocks on chromosome 3A (Table 2). The haploblock on 1B, spanning from 529.4 to 530.9 Mbp, includes four significant markers detected with BLUEs via BLINK, and two non-significant ones. The two haploblocks on chromosome 3A (located in the interval 510.8–528.5 Mbp) include mainly stable QTNs consistently detected in 2014 and 2017, along with markers evident only in 2017, and a non-significant one. The two haploblocks overlap by three SNPs (*AX-158523079*, *Excalibur_c29205_537*, and *Excalibur_rep_c76510_255*). The haploblock on chromosome 6A (interval 454.6–456.8 Mbp) consists of nine neighboring markers, most of which were detected by the MLM in 2014 and 2017, one marker was detected only in 2014, and another one was non-significant.

According to the LD decay information for each chromosome in the studied population [30], a genomic region of 1.5 to 3.0 Mbp up- and downstream of each stable SNP was considered to be a QTL. In summary, 25 QTL were defined as comprising either a stable QTN or a LD-haploblock, distributed across 14 chromosomes: 1A (1), 1B (1), 1D (1), 2A (3), 2D (1), 3A (4), 3B (1), 4A (2), 4B (1), 5A (1), 5B (3), 5D (1), 6A (4), and 7A (1) (Table 2). The phenotypic variance explained (PVE) by individual QTNs ranged between 0.0% and 11.4% (Table 2).

Notably, the common QTNs exhibited contrasting effects on PH, either reducing (spanning −5.3 to −16.1 cm) or increasing (3.7 to 18.6 cm). Among the stable SNPs, we specifically selected nine markers with relatively large effects, and PVE values distinct from zero. Haploblock *QPh.ippg-6A.3* was represented by seven out of nine markers, and haploblocks *QPh.ippg-3A.2* and *QPh.ippg-3A.3* by two markers each. Linear regressions considering the number of impactful QTNs demonstrated their pyramiding effect (*p* ≤ 0.001) (Figure 3). The correlation coefficients between the number of PH-reducing and PH-increasing alleles and the trait values were −0.326 and 0.457, respectively, at *p* < 0.001 (Figure 3).

Figure 4 illustrates the variation in PH based on the allele frequency at the selected loci. Genotypes with PH-decreasing alleles and those with alternative alleles exhibited phenotypic differences ranging from 6.1 to 18.3 cm. Notably, genotypes carrying the reducing allele “T” at the trait-associated locus *QPh.ippg-6A.1* on chromosome 6A (marker *Excalibur_c20597_569*) exhibited an average height reduction of 18.3 cm compared to those with the alternative allele “C.” Similarly, at the 4A locus (marker *CAP11_c3631_75*), accessions with the reducing allele “C” were 16.0 cm shorter than those with the alternative allele “T.” Additionally, accessions with trait-increasing alleles and alternative alleles showed PH differences spanning 6.9 to 19.6 cm, with the most significant disparity observed for marker *RFL_Contig5616_1779* on chromosome 5B. Marker loci on chromosome 3A seem mostly to have an increasing impact on PH. Five stable SNPs with PH-reducing effect (*AX-158543245*, *Tdurum_contig53138_302*, *AX-95228827*, *AX*-*94674935* and *AX-158527035*) and two with PH-promoting effect (*IAAV7384* and *BS00036878_51*) belong to haploblock *QPh.ippg-6A.3* (Table 2, Figure 4), suggesting that this region on chromosome 6A has an important impact on phenotypic variation.

Based on the above analysis, we selected the 14 most PH-influential alleles on chromosomes 3A, 4A, 5B, and 6A, and determined their frequency in accessions that were collected prior to the 1960s (historical varieties and landraces), as well as in the advanced modern varieties that were introduced between 1961 and 1990, and after 1990 (Appendix A). Distinct trends were identified—the frequency of all PH-reducing alleles increased, whereas the frequency of PH-promoting alleles decreased from landraces to contemporary releases.

Additionally, we ascertained the frequency of these alleles among the accessions originating from three production regions in the country (Appendix A). The alleles in haploblock *QPh.ippg-6A.3* showed a clear tendency—the PH-reducing alleles were notably more prevalent among accessions from the Southern region than in those from the Northern and Western regions, whereas the PH-increasing alleles were significantly less common. Allele frequency variations were noted in haploblocks *QPh.ippg-3A.2* and *QPh.ippg-3A.3* as well. Western accessions had PH-increasing alleles at two loci on chromosome 3A but showed a much lower frequency of the PH-reducing allele at marker locus *tplb0050h15_1287* compared to northern and southern accessions.

The genetic composition of important wheat varieties from each breeding period and the two primary production regions (Northern and Southern) was also evaluated (Appendix A). We compared the most widely grown varieties (100-10, Sadovo-1, Katya, Roussalka, Slavyanka, Enola, Aglika, Albena, and Todora), reference varieties for productivity (Sadovo-1 and Pryaspa), grain quality (Pobeda), drought tolerance (Katya), and nutrient efficiency (Fermer and Boryana), as well as accessions with extreme PH (No7, Slavyani, Rada). According to the analysis, the southern accessions are differentiated by alleles at all loci except for the *QPh.ippg-6A.1* locus. In contrast, the northern breeding materials exhibit variation primarily by alleles in the haploblocks on chromosome 3A, and by the loci on chromosomes 4A and 5B. The northern releases Charodejka, Roussalka, Albena, Aglika, and Todora have identical allele makeup at all 14 loci. Four of these genotypes share a common ancestor, S-13 (Appendix A). The historical southern variety, Slavyani, with its shorter culms and eight PH-reducing alleles but no PH-increasing alleles, differs significantly from other old accessions at 13 of the 14 loci. Interestingly, the modern southern variety, Boryana, shares the same allele composition. The allele makeup of Slavyani at loci on chromosomes 4A, 5B, and 6A is also present in the southern variety Sadovo-1, a long-term reference type for productivity. The nitrogen-efficient variety Fermer, created from Pobeda through mutagenesis, differs from its parent at two loci and resembles some old accessions. The variety Katya, internationally recognized for its drought resistance [31,32], has a distinct allele combination at loci on chromosome 6A, retaining some alleles from the old germplasm and the northern accessions. Among the important modern varieties, Rada has the highest PH BLUE value and is the only one carrying the most potent PH-increasing allele at locus *QPh.ippg-5B.3*, while lacking PH-reducing alleles, including the most widespread and effective one at locus *QPh.ippg-6A.1*.

### 2.3. Candidate Genes

A total of 15 genomic regions were screened to identify potential candidate genes using the wheat “Chinese Spring” reference genome IWGSC RefSeq v1.0. These regions were selected based on the associated SNPs with high effects on the trait and non-zero PVE. The support interval around each marker was extended 1.5 to 3.0 Mbp up- and downstream, guided by the LD decay information for the individual chromosomes [30]. For haplotype blocks, the search interval expanded to the first neighboring markers on both sides. We considered only genes with high confidence annotations as candidates potentially involved in controlling PH. In this context, we identified 668 candidate genes across the 15 scanned regions (Appendix A). Out of these, 21 genes contained a significant QTN or were very close to one. Among the remaining 647 genes, 97 were selected as relevant to processes affecting plant growth. According to their annotated functions, these genes fall into several categories: those related to the biosynthesis of plant hormones or signaling, genes affecting cell proliferation and elongation, genes implicated in nitrogen assimilation and photosynthesis, genes contributing to culm strengthening, stress-inducible genes, transcription factor-encoding genes and other regulatory elements.

The expression data of the selected pertinent candidate genes were obtained from the Wheat Expression Browser powered by expVIP in leaves/shoots at seedling, vegetative, and reproductive stages in several wheat varieties (https://www.wheat-expression.com/) (accessed on 18 July 2024). Most of the genes containing significant SNPs are well expressed (Figure 5a, Table 3). Out of the 97 selected candidate genes within the support intervals around significantly associated markers or within haploblocks, 31 had weak to very high expression levels (Figure 5b, Table 3). A particularly strong expression pattern was recorded for *TraesCS6A01G241400*, a gene on chromosome 6A hit by the marker *AX-158526868* (position: 452,959,034 bp) and verified by both multivariate methods, BLINK (in 2014 and 2017) and FarmCPU (in 2014) (Figure 2b and Figure 5a, Table 2). This gene is translated into a chaperone DnaJ protein (Table 3). Among the potential candidates in the defined QTL, gene *TraesCS2D01G500600* on chromosome 2D encoding glutamine synthetase showed exceptionally strong expression in all examined phenological phases (Figure 5b, Table 3). The most promising key genes involved in plant growth processes and thus affecting plant stature are discussed in the next section.

## 3. Discussion

Twenty years ago, 98% of the land under wheat cultivation in Bulgaria was sown with domestic varieties. Today, however, multinational seed companies increasingly address farmers’ interests in growing more profitable varieties. To remain competitive, Bulgarian breeders must adopt modern breeding technologies to develop a diverse range of wheat varieties, where high-yielding and high-quality genotypes complement each other. Currently, efforts are underway to restore farmers’ interest in Bulgarian-developed varieties that are nutrient-efficient, high-yielding under low-input agricultural practices, and tailored to the country’s specific climatic conditions [33,34,35,36].

Plant height is a crucial agronomic trait in wheat, significantly impacting both plant architecture and yield component traits [37]. Given the breeding focus on semi-dwarfism, identifying novel genetic variations related to wheat PH is essential for knowledge-based crop development. In this study, we explored phenotypic variation in PH across a population of modern semi-dwarf varieties and pre-“Green Revolution” tall accessions, including historical varieties and landraces. The results enhance our understanding of the genetic composition of improved wheat plants, laying the foundation for further advancements in wheat breeding in Southeastern Europe.

### 3.1. Phenotypic Variation

We observed a high degree of variability in the trait, influenced by both genotypic and environmental factors. This range of PH was notably narrower than that reported for other collections, including recent and historical accessions released before the 1960s, such as the Scandinavian germplasm [38] and the US germplasm [39]. It was still less variable than the PH variation observed in a panel of 319 Chinese accessions [40]. Analyzing accessions developed over nearly nine decades of breeding revealed that PH varied in correspondence with the year of release for different varieties (Appendix A). A pivotal moment in the genetic reduction in PH within the Bulgarian wheat collection occurred in the 1960s with the introduction of height-reducing genes [15]. Initially, the *Rht8* gene was transferred from the Italian variety “San Pastore”, a descendant of the Asian donor “Akakomugi”. Subsequently, other Italian, French, Serbian, Russian, and Ukrainian ancestors contributed to its presence. Despite its relatively moderate reducing effect on height, *Rht8* predominates in the Bulgarian wheat germplasm due to its longer coleoptile—an advantage in the early-season drought-prone environment characteristic of Southeastern Europe. Additionally, *Rht8* is tightly linked with the photoperiod insensitivity gene *Ppd-1*, which confers earlier heading, helping plants avoid high temperatures and drought stress [41]. The *Rht-B1d* allele (commonly known as the “Saitama-27” allele or *Rht1S*) has been suggested in the Bulgarian germplasm based on molecular data, GA-test results, and pedigree analysis [7,15]. It is often combined with *Rht8*. In contrast, the first introduced and most popular “Green Revolution” “Norin10”-derived alleles *Rht-B1b* and *Rht-D1b* occur much less frequently in this material [7,15].

### 3.2. Genomic Loci Significantly Associated with Plant Height in Wheat

To identify genomic loci involved in regulating PH within the studied wheat population, we employed a GWAS using three statistical models: a single-locus model (MLM) and two multi-locus models (FarmCPU and BLINK). The use of multiple statistical models is a common practice to enhance result robustness [42]. Given the complex genetic architecture of quantitative traits such as PH and their sensitivity to environmental factors, it is unsurprising that different models yield varying outcomes. In our study, all three models detected significant loci associated with PH. Specifically, the MLM identified significant QTNs in 2014 and 2017, BLINK detected QTNs in 2021 along with BLUE values, and the FarmCPU model revealed only a few significant SNPs in 2014.

Based on the Q–Q plots, the MLM showed significantly inflated *p*-values, indicating the potential effects of population stratification and cryptic relatedness among individuals in the association panel concerning the phenotype of interest (PH). Recent reports have highlighted distinct sub-populations and high relatedness within the Bulgarian germplasm [29,30], which is also evident from the genealogy data (Appendix A). To mitigate spurious positive signals, we focused on QTNs that surpassed the FDR threshold [−log_10_ (*p*-value) > FDR, adjusted *p*-value < 0.001]. For most environments, both the FarmCPU and BLINK Q–Q plots indicated that significant SNPs predominantly fell below the red line (null hypothesis), suggesting potential overfitting of these complex models and resulting in false negatives. However, in 2021 and BLUEs, BLINK exhibited deviations in *p*-values from the expected distribution, starting closer to the plot origin. This deviation could be attributed to population stratification but may also reflect the extremely polygenic nature of PH, and that the phenotypic variation likely arises from diverse biosynthesis and signaling pathways, and cellular processes [43].

According to a comprehensive review by Xu et al. [27] that summarizes two decades of research on the genetic regulation of PH, more than 600 genes, QTL, or GWAS loci associated with PH have been identified across all wheat chromosomes. In our study, we identified twenty-one environmentally stable QTNs that were detected in at least two environments or using multiple statistical models. Additionally, we considered four haplotype blocks as interesting loci for manipulating PH. These 25 loci were distributed across 14 wheat chromosomes. To validate their significance, we compared these loci to previously published genes, loci, or markers known to affect PH [27,38,44,45,46,47,48,49,50,51,52,53,54,55,56,57,58].

Our research did not reveal any associations between PH and the dwarfing alleles *Rht-B1b*, *Rht-D1b*, *Rht8*, and *Rht24* commonly used in worldwide wheat breeding. Similar findings were reported by Bellucci et al. [38] in an association mapping study of important agronomic traits, including PH, within Scandinavian wheat germplasm. In our study, we focused on the locus *QPh.ippg-4B* on chromosome 4B. This locus, marked by *wsnp_CAP12_rep_c4278_1949864* at position 552,018,200 bp, is notably distant from the *Rht-B1b* gene (*TraesCS4B02G043100*), which resides between 30,861,382 bp and 30,863,247 bp [53]. The substantial distance between these two loci suggests that different genes on chromosome 4B regulate PH. The lack of significant trait associations with markers *TG0010a* and *TG0010b* on chromosome 4B (position: 30,861,559 bp) (diagnostic for *Rht-B1b*, and present in the 25K Infinium iSelect array that was used for genotyping the panel) can be attributed to the limited presence of the *Rht-B1* alleles in our population [7]. Additionally, the *Rht*-*D1b* allele is extremely rare in the Bulgarian germplasm [7], and is probably absent in the current population. Despite a significant proportion of studied accessions carrying the GA-responsive gene *Rht8*, this gene was not detected using any of the three GWAS models. Similarly, the *Ppd-1* photoperiod sensitivity gene, which maps close to *Rht8* [12], remained undetected. Both *Rht8* and *Ppd-1* genes are located on the short arm of chromosome 2D, where marker polymorphism levels are very low [59]. Specifically, *Rht8* spans the interval 18.0–27.9 Mbp, while *Ppd-1* occupies the range of 32.8–40.0 Mbp [27]. The limited number of markers in the 25K SNP array within these intervals (15 for the 18.0–27.9 Mbp and 6 for the 32.8–40.0 Mbp) likely contributed to the failure to detect significant trait associations with *Rht8* and *Ppd-1*. Interestingly, the four QTL on chromosome 6A (*QPh.ippg-6A.1*, *QPh.ippg-6A.2*, *QPh.ippg-6A.3*, and *QPh.ippg-6A.4*) occupy more distal positions in both the short (6.7 Mbp) and long (453.0–514.5 Mbp) arms. These QTL differ from the well-known *Rht* genes on chromosome 6A, such as the allelic *Rht14/16/18* (9.2–85.3 Mbp; [60]), *Rht24* (411.9–414.9 Mbp; [23]), and *Rht25* (144.0–148.0 Mbp; [24]).

Instead, our GWAS revealed QTL that function independently of the known dwarfing genes, thus opening up new possibilities for the genetic regulation of PH. Our analysis confirmed some previously reported genomic regions. For example, the comparison with the QTL-rich clusters (QRC) associated with PH [27] showed that loci *QPh.ippg-4B* (position: 552.0 Mbp), *QPh.ippg-5A* (591.3 Mbp), and *QPh.ippg-5B.3* (692.6 Mbp) overlap with QRC 4B-IV (543.7–552.0 Mbp), 5A-VI (585.4–597.3 Mbp), and 5B-V (690.4–697.1 Mbp), respectively. The reported 5A-VI genomic region involves the major vernalization locus *Vrn1-5A*. Literature evidence suggests that vernalization may affect the duration of stem elongation [61]. A major locus for stem elongation has been mapped to the region where *Vrn1* resides on chromosome 5A [62]. However, *Vrn1*-*5A* is located at position 587,411,823–587,423,240 bp, which is 3.9 Mbp away from marker *Excalibur_c26671_57* (position: 591.3 Mbp) which in our study defines the locus *QPh.ippg-5A* (Table 2). The same marker is in proximity (distance 0.2–2.0 Mbp) to four described loci on chromosome 5A regulating PH [57], but far from the *Rht9* and *Rht12* mapped on the long arm of chromosome 5A [27,63]. Locus *QPh.ippg-5B.2* (marker *RAC875_rep_c111720_149*, position 526.6 Mbp) is included in the genomic interval 515.0–547.0 Mbp where marker–trait associations (MTAs) for grain yield are co-localized with PH loci as described by Horváth et al. [54]. Loci *QPh.ippg*-*2A.1* (marker *CAP12_c259_307*, position 15.9 Mbp) and *QPh.ippg-6A.1* (marker *Excalibur_c20597_569*, position 6.7 Mbp) coincide with PH-associated genomic regions reported by Tyrka et al. [58]. Loci *QPh.ippg-2A.3*, *QPh.ippg-5B.2*, and *QPh.ippg-7A* are in proximity to other MTAs on chromosomes 2A, 5B, and 7A, respectively, identified by QTL or GWAS analyses [27].

Thirteen stable QTNs and four haploblocks, located on chromosomes 1A, 1B, 1D, 2A, 2D, 3A, 3B, 4A, 5B, 5D, and 6A have not been previously associated with PH; hence, it is plausible that these seventeen loci represent novel findings. Among them, four loci *(QPh.ippg-1D*, *QPh.ippg-3B*, *QPh.ippg-4A.2*, and *QPh.ippg-6A.4*) exert a height reducing effect, and nine loci (*QPh.ippg-1A*, *QPh.ippg-2A.2*, *QPh.ippg-2D*, *QPh.ippg-3A.1*, *QPh.ippg-3A.4*, *QPh.ippg-4A.1*, *QPh.ippg-5B.1*, *QPh.ippg-5D*, and *QPh.ippg-6A.2*) have a height increasing effect, whereas the four haploblocks harbor SNPs with either reducing or increasing effect. Below, a few loci that show promise for breeding are addressed.

The relationship between PH and grain yield varies according to the severity of dwarfism and varietal background [64]. In general, optimum grain yields are achieved within a height range of 70–100 cm [64]. The analysis of the additive effect of PH-reducing and PH-increasing alleles indicated that increasing their number reduces or increases the height, respectively. Among the loci with the largest potential to modulate PH are those on chromosomes 3A, 4A, 5B, and 6A. Notably, the frequency of PH-reducing alleles at these loci has increased, while the frequency of PH-promoting alleles has decreased in the modern releases compared to the older germplasm. This trend demonstrates breeders’ adept manipulation of PH by integrating major semi-dwarfing genes with other loci affecting PH. Moreover, addressing the specific microclimate requirements of different regions within the country has resulted in different patterns in terms of spatial distribution of the PH-influencing alleles. Thus, the observed prevalence of PH-reducing alleles in the haploblock *QPh.ippg-6A.3* within the southern germplasm indicates a specific selection signature. This may be related to the need for further height reductions in southern genotypes, which are primarily *Rht8* gene carriers. Purposeful selective pressure has also altered the regions on chromosome 3A, encompassing haploblocks *QPh.ippg-3A.2* and *QPh.ippg-3A.3*, in both northern and southern subsets. Identifying these selection signatures is crucial for understanding the altered genetic makeup of plants, which is beneficial for successful crop breeding [65]. Most agronomic traits are influenced by genomic regions defined by underlying variation in SNP loci. These genomic regions are often co-inherited as haplotype blocks, which serve as effective units of selection for breeders [66]. Therefore, accurate marker-assisted selection applications necessitate the use of haplotypes composed of multiple SNPs, rather than utilizing single SNPs, in target regions [28]. The identified marker–trait associations have potential to benefit the work of wheat breeders not only in Bulgaria, but also in European regions of similar agrometeorological conditions.

### 3.3. Novel Potential Candidate Genes behind Plant Height in Wheat

A major motivation for conducting association mapping is to use the detected associations to determine the biological cause of heritable phenotypes and provide a basis for the development of innovative breeding strategies. In the present study, we mapped 25 QTL associated with PH, 21 of which were stable QTNs and 4 of which were haploblocks. Seventeen of them are potentially novel. Furthermore, we focused on 15 genomic regions with moderate to high expression levels of the identified high-confidence candidate genes that are related to plant growth (Table 3).

Numerous processes, such as the activity of plant hormones, reactions to environmental cues like stress and nutrient availability, and controls over the cell cycle and development, contribute to the regulation of plant growth. We selected 97 potential candidate genes within the LD-supported intervals of the significant QTNs and around the haploblocks. Analysis of gene expression profiles during seedling, vegetative, and reproductive growth periods (Figure 5) showed the important role of potential candidates for shaping plant stature and wheat improvement. These genes are assigned to several classes:

#### 3.3.1. Genes Affecting Hormone Biosynthesis and Signaling

Plant hormones, including GAs, brassinosteroids, cytokinins, auxin, ethylene, and others, cooperate to regulate many aspects of plant growth, development, and stress defense [67]. In this study, the region of haploblock *QPh.ippg-3A.3* on chromosome 3A was found to contain a gene that is translated into gibberellin 2-oxidase (GA2ox). GA oxidases (GAoxs) are key enzymes in GA biosynthesis [68,69]. Loss-of-function mutations in *GA20ox* genes and the increased expression of *GA2ox* genes have a dwarfing effect in Arabidopsis [70], rice [71], barley [72], wheat [44], and other species [4]. In wheat, overexpression of *GA2ox* genes has been found responsible for the dwarfing genes *Rht12* on chromosome 5A [73], *Rht14/16/18* [74] and *Rht24* on chromosome 6A [23]. The overexpression of *GA2ox* results in the removal of GA precursors, a reduction in bioactive GA1 content, and consequently stem shortening [23]. The *GA2ox* gene identified in the present study is a likely and promising candidate gene for manipulating plant stature. Another class of enzymes required for the synthesis of bioactive GAs in plants are cytochrome P450 monooxygenases (CYPs, P450s) that catalyze the three steps of the GA biosynthetic pathway from ent-kaurenoic acid to GA12 [75]. It has been proposed that *P450* genes may play a role in the regulation of PH in wheat [44]. Other *P450* genes underlying rice dwarfing genes are involved in brassinosteroid biosynthesis [76], or are associated with retardation of cell extension [77]. In our study, we found ten genes encoding cytochrome P450, of which six were located on chromosome 3A and exhibited no expression, while four were situated on chromosome 6A and displayed varying degrees of expression (Table 3 and Appendix A, Figure 5b).

Interestingly, the above-cited genomic region on chromosome 3A (*QPh.ippg-3A.3*) carries a gene encoding an auxin response factor (ARF). The output of our candidate gene search included another 15 genes coding for ARFs, mostly belonging to the SAUR (= small auxin-up RNAs) gene family on chromosome 2D (Appendix A). However, only the gene on chromosome 3A exhibited expression (Table 3, Figure 5b). ARFs constitute a large family of DNA-binding transcription factors that mediate the auxin action on the expression of multiple genes, thereby regulating every facet of plant growth and development [78,79]. In Arabidopsis, loss-of-function mutations in certain *ARF* genes are associated with distinct phenotypes, including effects on plant stature [80]. Two genes of moderate expression encoding ethylene responsive factor (ERF) were identified on chromosomes 3A and 6A (Table 3 and Appendix A, Figure 5b). The APETALA2/ERF (AP2/ERF) is a superfamily of transcription factors in plants that play a role in mediating plant responses to stress and environmental stimuli. In wheat, a novel *ERF* gene, *TaERF8* on chromosome 2B has been associated with PH and yield components [81].

Remarkably, on chromosome 6A, the QTN *Excalibur_c20597_569* resides within *TraesCS6A01G013600* (5.2–8.2 Mbp), a gene annotated as nitric oxide synthase (NOS) (Table 3), and displaying moderate to high expression (Figure 5a). Nitric oxide serves as an important signaling molecule throughout a plant’s lifespan and plays a key role in various developmental and stress response processes [82]. However, NOS-like activity has been reported only in certain algae and higher plants, with no homology to the mammalian NOS [82,83].

#### 3.3.2. Genes Affecting Cell Proliferation and Expansion

Stem elongation is a consequence of cell division occurring at the nodes. The early internode growth based on cell proliferation gradually gives way to growth based on cell expansion [84]. Here, we identified several genes with varying expression that are involved in the cell cycle, such as two genes in chromosomes 2D and 6A encoding cyclin protein—an activating subunit of cyclin-dependent kinases that are the driving forces of cell cycle progression [4]. Furthermore, two genes on chromosomes 2D and 3A that translate into a regulator of chromosome condensation (RCC1) protein were found. RCC1 protein is regarded as a critical cell cycle regulator that could be directly involved in the mitotic apparatus assembly and regulation of the cell cycle G1/S transition [85,86]. A gene of high expression in the 3A haploblock *QPh.ippg-3A.3* and directly linked to marker *wsnp_BE494474A_Ta_2_2* encodes katanin p60 ATPase-containing subunit. Katanin is a protein that affects the course of cell division and cell division plane orientation by regulating mitotic cytokinetic and cortical microtubule arrays. During plant growth, katanin also plays a role in cell elongation and morphogenesis [87]. An expansin-encoded gene was found in the region of *QPh.ippg-3B*. Expansins are cell wall proteins highly associated with wall loosening and cell size and shape, thus regulating cell wall extension during plant growth [88]. Modulation of stem elongation by GAs is partly achieved through expression of α-expansins [89]. In wheat, the growth of coleoptile has been closely related to the activity and expression of expansins [90]. We also found three other genes on chromosomes 4A and 6A translated into E3 ubiquitin-protein ligase and ubiquitin-conjugating enzyme E2, respectively (Table 3, Figure 5b). These proteins are components of the Destruction (D)-box proteolytic pathway thought to be related to another checkpoint of the cell cycle—the exit of mitosis [91].

#### 3.3.3. Genes Involved in Nutrient Assimilation and Photosynthesis

The “Green Revolution” *Rht* genes exert some drawbacks on agronomic traits, including lower NUE [92]. So, for semi-dwarf wheat varieties, improvement in NUE is required. This could be achieved by enhancing the activity of enzymes involved in nitrogen metabolism, among other possibilities [93]. An important player in nitrogen utilization and growth, and ultimately grain yield, is glutamine synthetase (GS), a key enzyme in the ammonium assimilation pathway. In our study, we detected a gene (*TraesCS2D01G500600* within locus *QPh.ippg-2D*) annotated as GS. This gene displays very high expression levels in leaves/shoots from the seedling to the reproductive stage (Table 3, Figure 5b). The chromosomal location of the gene suggests that it is more probably the plastid GS2 isoform, coded by genes on homoeologous group 2 chromosomes [94]. This study reported four favorable *TaGS2* haplotypes that may confer better seedling growth and improved N uptake during vegetative growth.

In addition to nitrogen assimilation, photosynthesis plays a crucial role in wheat growth and development. In our study, we identified a gene (*TraesCS3A01G041000*) associated with the significant marker *Kukri_c86406_100*, which has a high positive effect on PH (approximately 11–14 cm). This gene encodes a signal peptidase subunit family protein. Notably, in Arabidopsis, plastidic type I signal peptidase (Plsp1) is essential for the correct assembly of thylakoids, which are critical for driving photosynthesis. Disruption of *Plsp1* expression due to a mutation resulted in poor seedling growth, as observed in a study by Shipman-Roston et al. [95]. Although there are no specific studies on signal peptidase function in wheat growth, the above-cited findings [95] and the current data suggest its potential involvement in stem elongation.

#### 3.3.4. Genes Enhancing Strength of Wheat Stem

The structural carbohydrates in wheat stem cell walls, including cellulose, hemicellulose, and lignin, significantly influence stem mechanical strength and contribute to lodging resistance [96]. Our gene search identified three genes involved in the biosynthesis of cell wall components—two genes on chromosome 3A coding for cellulose synthase and vacuolar-processing enzyme, and one gene on chromosome 6A annotated as shikimate/quinate hydroxycinnamoyl transferase (Table 3, Figure 5). These findings align with earlier reports on QTL for stem strength mapped via linkage analysis on chromosomes 3A and 3B [97]. The cellulose synthase superfamily includes closely related cellulose synthase-like (Csl) proteins. Notably, a candidate gene associated with the wheat tiller inhibition gene (*tin*) was predicted to encode a Csl protein. This gene is linked to reduced tillering, stronger stems, and thicker lignified cell walls [98]. Additionally, the vacuolar processing enzyme gene *TaVPE3cB* has been identified as a candidate gene for a QTL related to wheat pith thickness on chromosome 3B [99]. Finally, hydroxycinnamoyl-CoA shikimate/quinate hydroxycinnamoyl transferase is among the key enzymes for regulating lignin biosynthesis and composition [96].

#### 3.3.5. Genes Implicated in Stress Responses

In our study, we identified several genes encoding proteins with defense functions in wheat. Most of these genes contain significant SNPs (Table 3). The products of these genes include the following: (1) Senescence/Dehydration-Associated Protein: Although the precise role of these proteins in wheat stem growth remains unknown, up-regulated expression of genes for senescence/dehydration-associated proteins and other drought-responsive genes has been linked to improved tolerance to minor drought stress in sensitive *Camellia* genotypes [100]; (2) Vascular Plant One Zinc Finger Transcription Factor (VOZ): VOZ transcription factors act as positive regulators of several salt-responsive genes [101]; (3) Glucan Endo-1,3-Beta-Glucosidase 3 (Gns): Some *Gns* genes encode pathogenesis-related proteins signifying plant defense reactions induced upon infections with pathogens [102]; (4) NBS-LRR-Like Resistance Protein: Proposed as a candidate gene for *Rht13* on chromosome 7B, a point mutation in the *Rht-B13b* allele autoactivates the NBS-LRR gene, resulting in a height reduction comparable to *Rht-B1b* and *Rht-D1b* [50]; (5) Chaperone DnaJ: These proteins have been implicated in response to diverse stresses [103,104,105].

Our findings align with the concept of growth–defense tradeoffs observed in plants. While deploying defense mechanisms is crucial for plant survival, it often comes at the expense of growth [67]. Interestingly, some of the SNPs associated with defense genes (*TraesCS3A01G291900*, *TraesCS4A01G053500*, and *TraesCS6A01G241400*) appear to increase PH, suggesting that they do not necessarily force a compromise between growth and defense.

#### 3.3.6. Genes Encoding Regulatory Elements

Plant growth regulation is orchestrated by a network of interacting signaling pathways, wherein transcription factors (TFs)—such as F-box, WRKY, bZIP, Myb, and TFIIB subunit 4b—and also methyltransferases and protein kinases play important roles. TFs serve as master regulators of gene expression, maintaining a delicate balance between growth and stress responses in plants [106]. In our study, we identified and examined the expression of several responsible genes (Table 3, Figure 5). Among these, three genes containing significant QTNs encode protein kinases. These proteins are fundamental players in plant hormone-mediated signaling, nutrient sensing, and cell cycle control [107]. Additionally, DNA and RNA methylation represent epigenetic and epitranscriptomic modifications that allow plants to adapt to both internal and external stimuli [108,109]. In this context, we report the discovery of a gene on chromosome 3A that codes for a dual-specificity RNA methyltransferase, and two genes on chromosome 6A, annotated as O-methyltransferase protein and cytosine-specific methyltransferase. Although the precise role of these proteins in wheat PH regulation remains incompletely understood, recent studies have highlighted connections between DNA and RNA methyltransferases and stress-related developmental processes [108,109].

## 4. Materials and Methods

### 4.1. Germplasm

A natural association population of 179 winter wheat (*Triticum aestivum* L.) accessions originating from Bulgaria was used as plant material in this study. The population represents the history of almost a century of wheat breeding in Bulgaria, and includes 131 semi-dwarf varieties, mainly released after the 1960s, and 48 pre-“Green Revolution” accessions (including old landraces and tall historical varieties that are no longer in production). The period of research missions to gather landraces’ seeds for gene bank collections [110] and the early breeding of now-historical types extended from 1925 to the 1960s, according to the currently available information (Appendix A). The old accessions were chosen based on collection site data to represent different geographical and eco-climatic zones in Bulgaria, and the seeds from this gene pool were derived from seed gene banks at the Leibniz Institute for Plant Genetics and Crop Research (IPK), Gatersleben, Germany, and the Crop Research Institute, Prague, Czech Republic. Seeds from the short-stemmed varieties were supplied by the two main wheat breeding centers in Bulgaria (Dobrudzha Agricultural Institute, General Toshevo, and the Institute of Plant Genetic Resources, Sadovo), as well as by private breeding companies. This set was previously analyzed at the genomic level using the 25K Infinium iSelect array (SGS Institut Fresenius GmbH TraitGenetics Section, Gatersleben, Germany). Previous studies covered the population structure, kinship estimation, and genetic diversity [29,30]. In brief, the population is composed of three distinct sub-populations, with the old genotypes forming a distinct group [29]. The heatmap of the values in the kinship matrix and the phylogenetic tree show high relatedness among the genotypes [30]. Information on the year of release, *Rht* status and known genealogy of the accessions is given in Appendix A. The occurrence of *Rht* alleles was postulated based on previous studies [7,15,111] using plant response tests to exogenous GA [112], DNA markers specific for *Rht-B1* and *Rht-D1* loci [113], and *Xgwm261* microsatellite locus, diagnostic for *Rht8* [12].

### 4.2. Experimental Design and Trait Measurements

To analyze natural phenotypic variation for PH, a field experiment was conducted in Sofia, Bulgaria (42°41′ N, 23°19′ E), during three crop seasons (2013/14, 2016/17, and 2020/21), denoted here by the year of harvest (2014, 2017, and 2021). Wheat accessions were grown in a random design as space-planted two-row plots (1 m row length, 5 cm between the plants within the row, 20 cm between the rows) with two replications in each growing season. The soil type at the experimental site is leached vertisol, pH 6.1, 3.1% humus, 1420 mg total N/kg soil, of which 18 mg inorganic N. Plants were supplied with 120 kg N/ha as ammonium nitrate in two split doses, 40 kg/ha two weeks after sowing and 80 kg/ha before stem extension. Standard agricultural measures were used to control the pests. The weather conditions for the experimental location during the three crop seasons are represented in Appendix A. In the periods spanning from October to July, the average monthly temperature fluctuated between −5.8 °C and 23.2 °C. During this time, monthly precipitation varied from 4.7 mm to 150.5 mm, resulting in an average annual precipitation of 682 mm—approximately 20% higher than the average for the location. Notably, in contrast to the abundant rainfall observed from April to July 2014, the precipitation levels in May and June 2017 were consistently below the climate average (Appendix A). This pattern suggests a sustained moderate drought during anthesis and grain filling in 2017. In 2021, a moderate drought during anthesis was observed, followed by a more severe final drought. At maturity, final PH data were collected from the main tillers of five representative and competitive plants taken from the center of the plots. This process was repeated for each replication, resulting in a total of ten plants per accession per growing season. PH was measured from the ground to the top of the ear excluding awns. The mean value of the measured culms represented the trait value of each accession in each growing season.

### 4.3. Field Experiment Statistical Analyses

Initially, for each accession, the package Genomic Association and Prediction Integrated Tool (GAPIT) v.3 [114] in R software v. 4.4 was employed to calculate the BLUE values for the trait values across the growing seasons to eliminate the environmental impact by assuming the genotype as a fixed effect and the growing season as a random effect.

The Shapiro–Wilk test using the web program MVApp (https://mvapp.kaust.edu.sa) (accessed on 10 April 2024) was performed to verify the normal distribution of the phenotypic data and BLUEs. The significant differences in PH among genotypes, crop seasons, and the genotype-by-environment interaction effects were tested by factorial ANOVA. The empirical phenotypic data collected over each growing season and the mean BLUE values were used to compute the Pearson correlation coefficients (*r*), and to estimate trait repeatability (*p*-value ≤ 0.05). Broad sense heritability *h*^2^ for PH was calculated with the following formula:$$h^{2}=\frac{\sigma_{G}^{2}}{\left(\sigma_{G}^{2}+\frac{\sigma_{E}^{2}}{nE}\right)}$$
where $\sigma_{G}^{2}$ is the genotype variance, $\sigma_{E}^{2}$ is the variance of the residual, and *nE* is the number of environments (growing seasons). To estimate heritability in each environment, the same formula was used, where *nE* in the denominator is the number of replications in a given environment.

All phenotypic data analyses were implemented with STATISTICA 14 [115].

### 4.4. GWAS and Haplotype Analysis

A GWAS of the wheat dataset of PH was performed with the empirical phenotypic data from three crop seasons, the calculated BLUE values, and the filtered set of 17,083 SNPs out of 19,019 polymorphic markers from already available genotypic data [29]. The population structure and the relatedness between the genotypes were considered prior to performing the marker–trait association analysis. Population structure was modeled using the Bayesian clustering algorithm in STRUCTURE 2.3.4 and described by Aleksandrov et al. in [29]. The membership coefficients (Q-values) were ascertained [29], and the kinship (K) was estimated using the method developed by VanRaden [116] as described by Kartseva et al. in [30]. For the GWAS, three models were employed. Initially, a mixed linear model (MLM) [117] including both population structure and kinship relations (MLM Q+K) as covariates to reduce the number of spurious signals was conducted in TASSEL v. 5. Aiming to control the false positive signals, the threshold of statistically significant QTNs was set at −log_10_ (*p*-value) > FDR (False Discovery Rate), at a significance level of adjusted *p*-value < 0.001. To calculate the adjusted *p*-value, the formula proposed by Benjamini and Hochberg [118] was used: adjusted *p*-value = *p*-value × (*m*/*k*), where *m* is the length of the *p*-value vector or the number of tests (in our case 17,083), and *k* is the rank of the SNP. FDR was computed for each environment.

Additionally, two state-of-the-art and powerful multivariate GWAS algorithms—the Fixed and random model Circulating Probability Unification (FarmCPU) and Bayesian-information and Linkage-disequilibrium Iteratively Nested Keyway (BLINK)—were applied using GAPIT v.3 in R. By combining the benefits of stepwise regression (fixed effect model) with the MLM, and mitigating their drawbacks, the FarmCPU model successfully controls the number of false positive signals, making this approach very accurate for GWASs [119]. Unlike FarmCPU, which assumes an even distribution of QTNs throughout the genome, in BLINK, this requirement is eliminated and LD information is used instead [120,121]. QTNs that were above the threshold of −log_10_ (*p*-value) > FDR with adjusted *p*-value (FDR) < 0.01 were considered statistically significant.

Additive effects and *R*^2^ (percent phenotypic variation) of QTNs were quantified in TASSEL v.5 and Excel 2016 (Microsoft, Redmond, WA, USA). To compute the percentage of phenotypic variation explained (PVE, in %) by each QTN (*R*^2^), the difference of *R*^2^ with and without the strongest linked SNP was used.

Pairwise LD (in Mbp) was determined in R for each chromosome separately and described previously [29]. The LD decay values varied from 1.5 to 3.0 Mb on the individual chromosomes [30]. For chromosome regions, where the QTNs formed clusters, the markers were grouped in LD blocks (haplotype blocks, haploblocks) if neighboring markers displayed strong pairwise LD (squared allele frequency, *r*^2^ > 0.5).

Trait-associated significant SNPs obtained with BLUEs and shared with at least one individual growing season, or detected in more than one individual growing season, or detected by more than one method were assumed to be environmentally stable QTNs.

GWAS results were visualized by Manhattan and Quantile–Quantile (Q–Q) plots generated using the R package qqman [122]. To determine whether there were false positives or negatives, Q–Q plots for the three models (MLM, FarmCPU, and BLINK) were compared.

### 4.5. Candidate Gene Nomination and Expression Analysis

For selected significant QTNs with high reducing or promoting effects on PH, and PVE > 0.0%, the flanking regions within the LD-estimated window on either side of the marker were explored for putative candidate genes. For significant markers clustered in haploblocks, all genes within such specific LD-blocks were considered for candidate gene prediction, whereby the search intervals were extended to the left and right neighboring marker using the markers’ information in the 25K genotyping array. The candidate genes were retrieved from the annotation of the wheat variety “Chinese Spring” reference genome IWGSC RefSeq v1.0 [123] from the Persephone database (https://web.persephonesoft.com/?data) (accessed on 15 May 2024).

The expression patterns of selected candidate genes were examined in the leaves/shoots at three phenological stages relative to stem growth (seedling, vegetative, and reproductive) under non-stress and stress conditions, and in several wheat varieties by utilizing the published wheat RNA-seq expression database available on the Wheat Expression Browser powered by expVIP (https://www.wheat-expression.com/) (accessed on 18 July 2024) [124,125]. Transcripts per million (tpm) was used to represent the gene expression levels.

## 5. Conclusions

In this study, a collection of 179 accessions representing almost a century of wheat breeding in Bulgaria was evaluated for variation in PH, and the underlying genetic factors were determined. GWAS results identified QTL that were independent of all known *Rht* genes, thereby providing new resources for the genetic manipulation of PH. While some of the detected QTL confirm previously reported ones, other loci appear to be novel and promising for breeding varieties with a balanced height–yield relation. Based on genes’ functional annotations, a selection of highly probable candidates was made among the numerous high-confidence genes recovered from the trait-associated genomic regions. Good candidates include genes involved in hormones’ biosynthesis and signaling, cell cycle, nitrogen metabolism, and photosynthesis, as well as genes implicated in plant stress defense, stem physical strength, and a number of master regulators of plant growth. Exploitation of new gene resources contributing to the control of valuable agronomic traits such as PH may facilitate the breeding of modern wheat varieties through molecular approaches.

## Figures and Tables

**Figure 1 plants-13-02775-f001:**
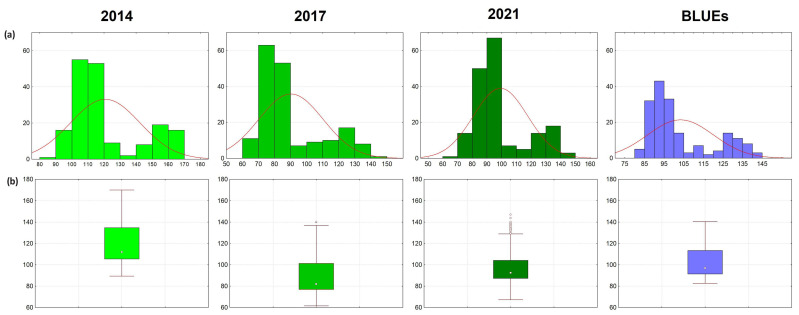
Frequency distribution (**a**) and boxplots (**b**) for plant height (in cm) in 179 wheat accessions from Bulgaria in three crop seasons and with the best linear unbiased estimator (BLUE) values. The boxplots show the median as a dot and the 1st and 3rd quartile as a box; whiskers depict the non-outlier range.

**Figure 2 plants-13-02775-f002:**
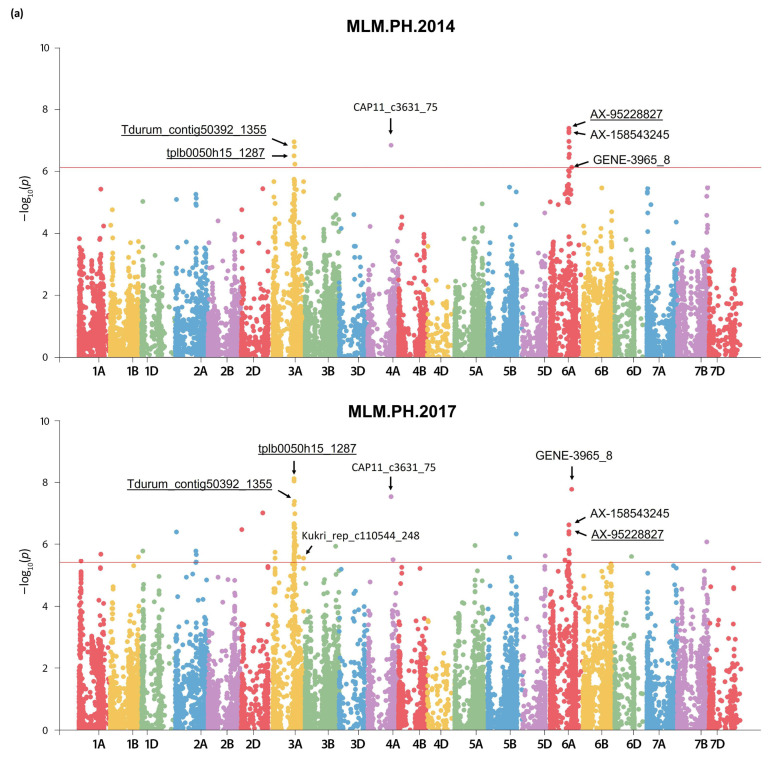
Manhattan plots for genome-wide associated scan of plant height (PH) in 179 Bulgarian wheat accessions, detected by (**a**) a single-locus model, and (**b**) two multivariate models. The colored horizontal lines indicate the significance threshold at −log_10_ (*p*-value) = FDR (False Discovery Rate), at a significance level of adjusted *p*-value < 0.001 (in (**a**)) and *p*-value < 0.01 (in (**b**)), computed for each environment. Arrows point to the novel loci associated with PH. In haplotype blocks on chromosomes 1B, 3A, and 6A, only peak SNPs are marked and underlined (see also Table 2).

**Figure 3 plants-13-02775-f003:**
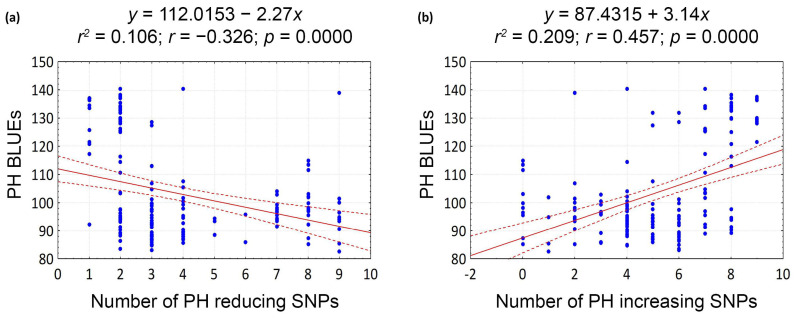
Scatter plots showing linear regressions (solid red lines) and 95% confidence intervals of the regressions (dashed red lines) of nine selected SNPs with (**a**) high reducing and (**b**) high increasing effects on plant height (PH) with PH BLUE values in 179 wheat accessions. The blue dots represent the PH BLUE values of accessions with corresponding number of SNPs.

**Figure 4 plants-13-02775-f004:**
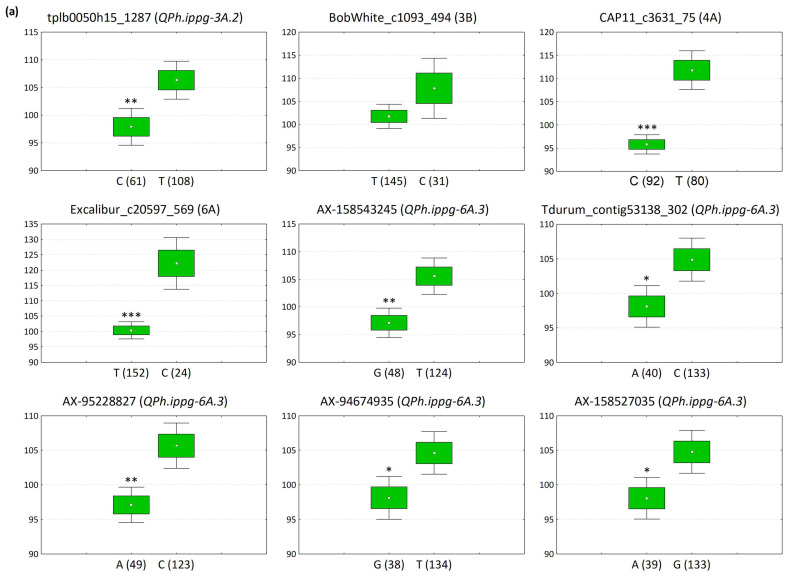
Allelic effects on plant height (PH) phenotype showing differences in PH BLUE values between accessions with PH-affecting alleles (left) and accessions with alternative alleles (right) at nine selected stable SNP loci. (**a**) PH-reducing SNPs; (**b**) PH-increasing SNPs. In parentheses, the full name of a QTL is noted for the SNPs belonging to a haplotype block; for the remaining SNPs, only the chromosome is annotated. The boxplots show the mean as a dot and the mean ± SE (standard error) as a box; whiskers depict the mean ± 1.96SE range. *, **, *** indicate significant difference at *p* ≤ 0.05, 0.01, 0.001, respectively, as calculated by Student’s *t* test.

**Figure 5 plants-13-02775-f005:**
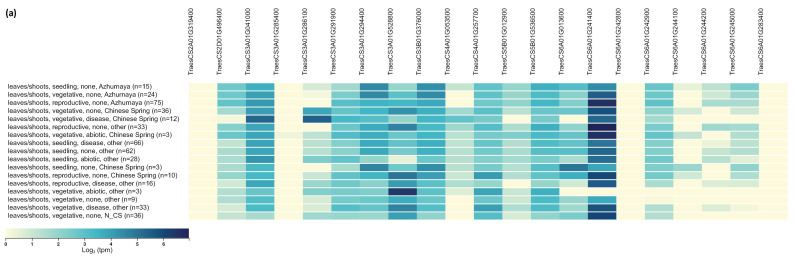
Heatmap showing gene expression levels (log_2_ tpm, transcripts per million) of selected putative genes controlling plant height in wheat: (**a**) genes containing significant SNPs or very close to them; (**b**) genes in the support interval around significant SNPs and haplotype blocks.

**Table 1 plants-13-02775-t001:** Summary statistics for plant height (PH) in 179 Bulgarian wheat accessions evaluated under three growing seasons (environments, Env.). Std. Dev., standard deviation; Std. Err., standard error; CV, coefficient of variation; *h*^2^, broad-sense heritability; BLUE, best linear unbiased estimator.

Env.	2014	2017	2021	Average	BLUE
Mean (cm) *	120.5 a	90.0 c	99.0 b	103.2	103.2
Min (cm)	89.4	61.5	67.3	77.0	82.6
Max (cm)	170.0	140.2	147.0	148.4	140.5
Std. Dev.	21.56	19.93	18.33	19.46	16.73
Std. Err.	1.61	1.49	1.37	1.45	1.25
CV (%)	17.9	22.1	18.5	18.9	16.2
*h* ^2^	0.89	0.97	0.77	0.98	

* Different letters denote significant differences between the mean values of PH among environments at *p* ≤ 0.001.

**Table 2 plants-13-02775-t002:** Significant loci controlling plant height in a population of 179 Bulgarian wheat accessions.

Chr.	QTL ^a^	SNP	Allele ^b^	Position (bp)	−log(*p*)	Effect (cm)	PVE (%) ^c^	Model_Environment
1A	*QPh.ippg-1A*	*AX-94907052*	T/G	12819799	7.2; 6.3	8.5; 7.0	0.3; 0.0	BLINK_2021; BLUE
1B	* QPh.ippg-1B *	*AX-158521149*	A/T	529395106				Not significant
		*BS00036362_51*	A/G	530478430				Not significant
		*BS00000066_51*	T/C	530480842	10.8	−15.0	0.0	BLINK_BLUE
		*wsnp_BG606586B_Ta_2_13*	A/G	530480991	5.6	−9.1	0.0	BLINK_BLUE
		*BS00010130_51*	A/G	530959498	9.1	14.1	0.0	BLINK_BLUE
		*RAC875_s110519_303*	A/G	530959790	10.0	−14.3	1.5	BLINK_BLUE
1D	*QPh.ippg-1D*	*AX-158571058*	T/C	303309200	12.2; 11.0	−16.1; −13.6	0.0; 0.0	BLINK_2021; BLUE
2A	*QPh.ippg-2A.1*	*CAP12_c259_307*	T/C	15875857	10.8; 6.1	18.3; 11.6	0.0; 0.0	BLINK_2021; BLUE
	*QPh.ippg-2A.2*	*Kukri_c29592_559*	T/G	545406692	20.5; 17.9	18.6; 15.6	6.7; 6.8	BLINK_2021; BLUE
	*QPh.ippg-2A.3*	*BobWhite_c17783_174*	A/C	744820620	9.4; 6.3	13.8; 9.7	0.8; 0.0	BLINK_2021; BLUE
2D	*QPh.ippg-2D*	*AX-94425197*	A/G	592783707	13.6; 10.3	16.9; 12.9	1.3; 2.8	BLINK_2021; BLUE
3A	*QPh.ippg-3A.1*	*Kukri_c86406_100*	T/C	21374604	10.5; 8.0	14.2; 10.9	5.9; 1.9	BLINK_2021; BLUE
	* QPh.ippg-3A.2 *	*IAAV3488*	A/C	510828578	6.6	−10.1	6.8	MLM_2017
		*wsnp_Ex_c10667_17387885*	A/G	514109963	7.0; 8.0	11.1; 11.7	7.0; 8.6	MLM_2014; 2017
		*wsnp_Ex_c2148_4035913*	A/G	514311380	5.5	9.1	5.5	MLM_2017
		*wsnp_Ex_c32003_40728918*	A/G	514313603	5.4	−8.9	5.4	MLM_2017
		*tplb0050h15_1287*	C/T	514494005	6.5; 8.1	−10.7; −11.8	6.5; 8.7	MLM_2014; 2017
	* QPh.ippg−3A.3 *	*AX-158523079*	C/G	514560784	7.3	−11.0	7.7	MLM_2017
		*Excalibur_c29205_537*	A/G	514623977	5.5	−9.1	5.4	MLM_2017
		*Excalibur_rep_c76510_255*	C/T	515889945				Not significant
		*Tdurum_contig50392_1355*	A/G	522060474	6.8; 7.4	11.2; 11.3	6.9; 8.0	MLM_2014; 2017
		*wsnp_Ex_rep_c69752_68711460*	A/C	523556188	5.9	10.1	6.1	MLM_2017
		*Ex_c18484_2048*	A/C	525200865	6.0	−10.1	6.1	MLM_2017
		*Ex_c18484_2026*	C/T	525200887	5.8	9.8	5.9	MLM_2017
		*AX-110568350*	A/G	526863833	6.6	−10.5	6.9	MLM_2017
		*wsnp_BE494474A_Ta_2_2*	G/T	528521643	6.2; 6.3	10.9; 10.6	6.1; 6.5	MLM_2014; 2017
	*QPh.ippg-3A.4*	*Kukri_rep_c110544_248*	A/G	742613091	5.5; 6.5	9.9; −5.3	5.5; 2.2	MLM_2017; FarmCPU_2014
3B	*QPh.ippg-3B*	*BobWhite_c1093_494*	T/C	591458339	6.1; 7.0	−7.9; −7.7	9.3; 0.0	BLINK_2021; BLUE
4A	*QPh.ippg-4A.1*	*AX-95188776*	A/G	45142425	7.1; 5.5	11.5; 8.8	9.3; 3.1	BLINK_2021; BLUE
	*QPh.ippg-4A.2*	*CAP11_c3631_75*	C/T	570466352	6.9; 7.5	−11.7; −11.9	6.9; 7.9	MLM_2014; 2017
4B	*QPh.ippg-4B*	*wsnp_CAP12_rep_c4278_1949864*	A/C	552018200	6.4; 7.8	11.5; 11.7	0.0; 0.0	BLINK_2021; BLUE
5A	*QPh.ippg-5A*	*Excalibur_c26671_57*	T/C	591319170	7.2; 6.4	−6.6; −5.5	0.1; 0.0	BLINK_2021; BLUE
5B	*QPh.ippg-5B.1*	*AX-108742604*	A/G	12860184	5.6; 7.0	7.7; 7.9	3.4; 4.6	BLINK_2021; BLUE
	*QPh.ippg-5B.2*	*RAC875_rep_c111720_149*	T/C	526590533	7.2; 10.2	11.2; 12.4	0.0; 1.9	BLINK_2021; BLUE
	*QPh.ippg-5B.3*	*RFL_Contig5616_1779*	C/T	692564868	6.3; 7.4	11.1; 8.0	6.5; 4.4	MLM_2017; BLUE
5D	*QPh.ippg-5D*	*AX-95147884*	A/G	238666906	12.6; 5.8	16.4; 9.2	0.0; 0.0	BLINK_2021; BLUE
6A	*QPh.ippg-6A.1*	*Excalibur_c20597_569*	T/C	6725084	6.5; 13.0	−9.3; −12.6	0.0; 1.3	BLINK_2021; BLUE
	*QPh.ippg-6A.2*	*AX-158526868*	T/C	452959034	8.0; 7.4; 7.0	4.6; 3.7; 6.2	11.4; 11.0; 5.2	BLINK_2014; 2017; FarmCPU_2014
	* QPh.ippg-6A.3 *	*AX-158543245*	G/T	454638223	7.3; 6.6	−12.8; −11.5	7.4; 6.8	MLM_2014; 2017
		*Tdurum_contig53138_302*	A/C	454649350	7.0; 5.8	−13.1; −11.2	7.1; 5.8	MLM_2014; 2017
		*IAAV7384*	G/T	454649422	7.3; 6.3	13.2; 11.6	7.4; 6.4	MLM_2014; 2017
		*AX-95228827*	A/C	454655948	7.4; 6.4	−12.9; −11.3	7.5; 6.6	MLM_2014; 2017
		*BS00036878_51*	A/G	455646736	7.3; 6.4	13.5; 12.0	7.4; 6.6	MLM_2014; 2017
		*AX-94674935*	G/T	455658391	6.5; 5.4	−12.6; −10.9	6.4; 5.4	MLM_2014; 2017
		*AX-158527035*	A/G	456601664	6.8; 5.7	−13.0; −11.2	6.8; 5.7	MLM_2014; 2017
		*AX-158587929*	A/G	456751529	6.6	12.3	6.5	MLM_2014
		*IACX2250*	A/G	456752252				Not significant
	*QPh.ippg-6A.4*	*GENE-3965_8*	G/T	514496178	6.1; 7.8	−10.4; −11.5	6.0; 8.3	MLM_2014; 2017
7A	*QPh.ippg-7A*	*AX-94599642*	A/G	675113805	9.9; 6.2	10.7; 7.2	0.4; 0.0	BLINK_2021; BLUE

^a^ Quantitative trait loci (QTL) represent the flanking regions within the linkage disequilibrium-estimated window on either side of the significant SNP. The haplotype blocks (underlined QTL) include clusters of significant SNPs and the flanking regions extended to the first neighboring SNPs in the genotyping array; ^b^ underlined allele = minor allele; ^c^ PVE = phenotypic variance explained.

**Table 3 plants-13-02775-t003:** Candidate genes with varying expression levels in selected genomic regions associated with plant height as revealed by GWAS.

QTL ^a^	Interval (Mbp)	SNP	Gene ID ^b^	Annotation	Expression
*QPh.ippg-2A.2*	543. 9–546.9	*Kukri_c29592_559*	* TraesCS2A01G319400 *	Zinc finger protein	No
*QPh.ippg-2D*	590.2–595.4	*AX-94425197*	* TraesCS2D01G496400 *	Cyclin-related family protein	Yes
			*TraesCS2D01G492700*	Regulator of chromosome condensation	Yes
			*TraesCS2D01G500600*	Glutamine synthetase	Yes
*QPh.ippg-3A.1*	19.4–23.4	*Kukri_c86406_100*	* TraesCS3A01G041000 *	Signal peptidase subunit family protein	Yes
			*TraesCS3A01G041200*	Ethylene-responsive transcription factor	Yes
* QPh.ippg-3A.2 *	510.7–516.3	*wsnp_Ex_c10667_17387885 tplb0050h15_1287*	*TraesCS3A01G282200*	F-box family protein	Yes
			*TraesCS3A01G284800*	Regulator of chromosome condensation	Yes
			* TraesCS3A01G285400 *	Transducin/WD40 repeat-like protein	No
			* TraesCS3A01G286100 *	Senescence/dehydration-associated protein-like protein	Yes
* QPh.ippg-3A.3 *	514.5–528.5	*Tdurum_contig50392_1355 wsnp_BE494474A_Ta_2_2*	*TraesCS3A01G289700*	WRKY transcription factor	Yes
			*TraesCS3A01G289900*	Cellulose synthase	Yes
			*TraesCS3A01G291000*	F-box family protein	Yes
			*TraesCS3A01G291600*	Elongation factor 4	Yes
			* TraesCS3A01G291900 *	Vascular plant one zinc finger TF	Yes
			*TraesCS3A01G292200*	F-box protein SKIP8	Yes
			*TraesCS3A01G292400*	Auxin response factor	Yes
			*TraesCS3A01G293700*	BZIP transcription factor	Yes
			*TraesCS3A01G294000*	Gibberellin 2-oxidase	Yes
			* TraesCS3A01G294400 *	Katanin p60 ATPase-containing subunit 4b	Yes
*QPh.ippg-3A.4*	740.6–744.6	*Kukri_rep_c110544_248*	* TraesCS3A01G528800 *	Vacuolar-processing enzyme	Yes
			*TraesCS3A01G530100*	Dual-specificity RNA methyltransferase	Yes
*QPh.ippg-3B*	589.7–593.3	*BobWhite_c1093_494*	* TraesCS3B01G376000 *	Methionine-tRNA ligase, putative	Yes
			*TraesCS3B01G376800*	Expansin	Yes
*QPh.ippg-4A.1*	43.1–47.1	*AX-95188776*	*TraesCS4A01G053400*	E3 ubiquitin-protein ligase	Yes
			* TraesCS4A01G053500 *	Glucan endo-1,3-beta-glucosidase 3	Yes
*QPh.ippg-4A.2*	568.5–572.5	*CAP11_c3631_75*	*TraesCS4A01G257500*	myb-like protein X	Yes
			* TraesCS4A01G257700 *	Inositol-tetrakisphosphate 1-kinase	Yes
*QPh.ippg-5B.1*	11.4–14.4	*AX-108742604*	* TraesCS5B01G012900 *	WD-repeat protein, putative	Yes
			*TraesCS5B01G013700*	F-box protein	Yes
*QPh.ippg-5B.3*	691.1–694.1	*RFL_Contig5616_1779*	*TraesCS5B01G535100*	NBS-LRR-like resistance protein	Yes
			* TraesCS5B01G536500 *	Kinase-like protein	Yes
			*TraesCS5B01G536600*	Myb-like protein X	Yes
*QPh.ippg-6A.1*	5.2–8.2	*Excalibur_c20597_569*	* TraesCS6A01G013600 *	Nitric oxide synthase 1	Yes
			*TraesCS6A01G012100*	Cytochrome P450	Yes
			*TraesCS6A01G012300*	Cytochrome P450	Yes
			*TraesCS6A01G012400*	Cytochrome P450	Yes
			*TraesCS6A01G015400*	O-methyltransferase family protein	Yes
			*TraesCS6A01G015800*	Cytosine-specific methyltransferase	Yes
			*TraesCS6A01G016200*	Ubiquitin-conjugating enzyme E2, putative	Yes
*QPh.ippg-6A.2*	451.5–454.5	*AX-158526868*	* TraesCS6A01G241400 *	Chaperone protein dnaJ-like	Yes
			*TraesCS6A01G241800*	Cyclin-like	Yes
			*TraesCS6A01G242600*	Shikimate/quinate hydroxycinnamoyl transferase	Yes
* QPh.ippg-6A.3 *	453.9–457.4	*AX-158543245* *Tdurum_contig53138_302* *IAAV7384* *AX-95228827* *BS00036878_51* *AX-94674935* *AX-158527035*	* TraesCS6A01G242800 *	Kinase family protein	No
			* TraesCS6A01G242900 *	Transcription initiation factor TFIID subunit	Yes
			*TraesCS6A01G243300*	Ethylene-responsive transcription factor	Yes
			* TraesCS6A01G244100 *	Protein kinase	Yes
			* TraesCS6A01G244200 *	Retrovirus-related Pol polyprotein from transposon TNT 1-94	Yes
			*TraesCS6A01G244900*	Cytochrome P450	Yes
			* TraesCS6A01G245000 *	Protein kinase	Yes
			*TraesCS6A01G284200*	Ubiquitin-conjugating enzyme E2, putative	Yes
*QPh.ippg-6A.4*	513.0–516.0	*GENE-3965_8*	* TraesCS6A01G283400 *	Eukaryotic aspartyl protease family protein	No

^a^ Quantitative trait loci (QTL) represent the flanking regions within the linkage disequilibrium-estimated window on either side of the significant SNP. The haplotype blocks (underlined QTL) include clusters of significant SNPs and the flanking regions extended to the first neighboring SNPs in the genotyping array; ^b^ underlined genes include the significant stable SNPs in the order they are listed in the column to the left; gene *TraesCS6A01G242900* (*QPh.ippg-6A.3*) contains three SNPs.

## Data Availability

Data are contained within the article and its Appendix A.

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
