# Peer review of "Exploring Novel Genomic Loci and Candidate Genes Associated with Plant Height in Bulgarian Bread Wheat via Multi-Model GWAS"

_plants, 2024, doi:10.3390/plants13192775_

Round 1

Reviewer 1 Report

Comments and Suggestions for Authors

The article is acceptable for publishing except for a couple of things:

1. The distributions of the genotypes by plant height seem very much skewed to the left (Figure 1a), in spite of the insistence of the authors that the Shapiro-Wilk test showed good significance levels for normality (p 0.05). As the raw data is not available, it is difficult to check the exact reason for that, but this should be clarified.
2. Line numbers 141 and 142 are overlaid on the Table 1.

Reviewer 2 Report

Comments and Suggestions for Authors

I think the title is too broad, it looks you have identified some novel genes or regulators controlling plant height, but actually, there is none.

 Before GWAS analysis, it should provide the genotyping results, population structure, and linkage disequilibrium of the materials.

 You have carried out GWAS using three methods: MLM, FarmCPU and BLINK, how to compare the results derived from different methods, which one is most powerful? You should provide data to describe why you chose the MIM results for further analysis. In my experience, FarmCPU works more powerful and can detect the SNPs with reasonably accurate.

 You have identified some candidate genes for loci identified by GWAS, but have you carried out candidate gene based LD and haplotype analysis for those candidate genes?

L124: at the beginning of the result part, it is better to describe what growing seasons (years) were the materials planted, and how the traits were measured, that will make it easy to read and understand the manuscript.

Reviewer 3 Report

Comments and Suggestions for Authors

Review of the manuscript titled: “Novel genetic regulators of plant height in bread wheat”.

The study is devoted to an important subject of studying the genetic control of plant height in Bulgarian wheats. The study methodology is relevant, the material used covered all the genetic variation  of Bulgarian wheat, the results are significant, contribute to our knowledge of the subject and deserve publication. The paper is also well written and easy to read. However, as any paper, there are avenues for improvement.

1.        Introduction. Since the study focuses on Bulgarian wheat, it make sense to add a short paragraph on wheat production in the country mentioning area, production regions, productivity dynamics and breeding framework.

2.        Supplementary table 1 presenting the material can be arranged by the period of collection so that old, intermediate and relatively new cultivars come together with average for plant height for each group.

3.        Discussion is perhaps a weaker section of the paper in some ways repeating the results. The reference to tables and figures in discussion is hardly justified. Broader discussion topics need to cover: a) the current breeding framework in Bulgaria and any differences in the breeding programs for PH related genes; b) Is there a tendency of foreign varieties being grown in Bulgaria on a growing scale, their origin and possible genetic make-up for PH and how the current study can help Bulgarian breeders to stay more competitive; c) Potential of the new markers use outside of Bulgaria; d) selection signature in respect to the key genes/markers identified in the study; e) Differences between the regions; f) the genetic make-up of the most commonly grown landmark varieties.

4.        The paper lacks the linkage between the genes and the genotypes. It would be well justified to add a table with 5-8 representative varieties from each breeding period, provide the PH BLUEs and the presence of the alleles for the key genes identified in the paper. The “representative” varieties can be most commonly grown or the extremes in the PH or any other sensible criteria. This will connect varieties and genes. If the authors want – they can also add frequency of the same key genes in the germplasm from different breeding period.

Round 2

Reviewer 2 Report

Comments and Suggestions for Authors

authors have modifed the manuscript properly according to the reviewer's comments.